# Deep Sequencing Reveals Central Nervous System Compartmentalization in Multiple Transmitted/Founder Virus Acute HIV-1 Infection

**DOI:** 10.3390/cells8080902

**Published:** 2019-08-15

**Authors:** Sodsai Tovanabutra, Rujipas Sirijatuphat, Phuc T. Pham, Lydia Bonar, Elizabeth A. Harbolick, Meera Bose, Hongshuo Song, David Chang, Celina Oropeza, Anne Marie O’Sullivan, Joyce Balinang, Eugene Kroon, Donn J. Colby, Carlo Sacdalan, Joanna Hellmuth, Phillip Chan, Peeriya Prueksakaew, Suteeraporn Pinyakorn, Linda L. Jagodzinski, Duanghathai Sutthichom, Suwanna Pattamaswin, Mark de Souza, Robert A. Gramzinski, Jerome H. Kim, Nelson L. Michael, Merlin L. Robb, Nittaya Phanuphak, Jintanat Ananworanich, Victor Valcour, Gustavo H. Kijak, Eric Sanders-Buell, Serena Spudich

**Affiliations:** 1U.S. Military HIV Research Program, Walter Reed Army Institute of Research, Silver Spring, MD 20910, USA; 2The Henry M. Jackson Foundation for the Advancement of Military Medicine, Bethesda, MD 20817, USA; 3Department of Medicine, Faculty of Medicine Siriraj Hospital, Mahidol University, Bangkok 10700, Thailand; 4SEARCH, Thai Red Cross AIDS Research Centre, Bangkok 10330, Thailand; 5Memory and Aging Center, Department of Neurology, University of California, San Francisco, CA 94158, USA; 6International Vaccine Institute, Seoul 08826, Korea; 7Academic Medical Center, Department of Global Health, University of Amsterdam, 1105AZ Amsterdam, The Netherlands; 8Department of Neurology, Yale University; New Haven, CT 06510, USA

**Keywords:** HIV-1, central nervous system (CNS), cerebrospinal fluid (CSF), compartmentalization, acute HIV-1 infection (AHI), transmitted/founder (T/F) virus, multiple infections, single-genome amplification (SGA), next-generation sequencing (NGS)

## Abstract

HIV-1 disseminates to a broad range of tissue compartments during acute HIV-1 infection (AHI). The central nervous system (CNS) can serve as an early and persistent site of viral replication, which poses a potential challenge for HIV-1 remission strategies that target the HIV reservoir. CNS compartmentalization is a key feature of HIV-1 neuropathogenesis. Thus far, the timing of how early CNS compartmentalization develops after infection is unknown. We examined whether HIV-1 transmitted/founder (T/F) viruses differ between CNS and blood during AHI using single-genome sequencing of envelope gene and further examined subregions in *pol* and *env* using next-generation sequencing in paired plasma and cerebrospinal fluid (CSF) from 18 individuals. Different proportions of mostly minor variants were found in six of the eight multiple T/F-infected individuals, indicating enrichment of some variants in CSF that may lead to significant compartmentalization in the later stages of infection. This study provides evidence for the first time that HIV-1 compartmentalization in the CNS can occur within days of HIV-1 exposure in multiple T/F infections. Further understanding of factors that determine enrichment of T/F variants in the CNS, as well as potential long-term implications of these findings for persistence of HIV-1 reservoirs and neurological impairment in HIV, is needed.

## 1. Introduction

Although acute HIV-1 infection (AHI) has been primarily studied for its role in the establishment of a persistent systemic infection with immediate as well as chronic effects on the immune system, HIV-1 is also known to disseminate to diverse tissue sites during the earliest stages of infection [1,2]. Early tissue dissemination may facilitate the formation of sites of viral persistence as tissues outside of the well-characterized circulating CD4^+^ T lymphocyte population are subject to distinct immune surveillance [3]. Additionally, differences in infected cell types and varying tissue penetration of antiretroviral therapy (ART) drugs may lead to incomplete viral suppression or ongoing immune perturbation, even after effective ART is initiated [4,5]. Understanding the establishment and persistence of HIV-1 at these tissue sites is critical for current efforts aimed at HIV-1 remission, as evidence of HIV-1 replication in sites other than systemic T lymphoid cells may be a potential source of HIV-1 despite peripheral T cell sources of HIV-1 replication being contained or eradicated.

The central nervous system (CNS) compartment is one of several sites (including breast milk, lung, and the genital compartment) in which compartmentalized HIV-1 replication has been observed in humans [6]. Most studies assessing viral compartmentalization in the CNS via cerebrospinal fluid (CSF) collection have been in individuals with chronic HIV-1 infection [7]. Previous studies have revealed that genital compartmentalization of HIV-1 can occur at the early stage of HIV-1 infection in humans [8], and compartmentalization in the anogenital tract has been described very early in rhesus monkeys during acute simian immunodeficiency virus (SIV) infection [9]. However, information about tissue compartmentalization of HIV-1 during acute infection is still limited.

Although HIV-1 does not infect neurons, it is known to be directly neuropathogenic through local viral infection of other cells within the CNS and associated immunopathogenesis. Disease manifestations may include meningoencephalitis and clinical HIV-associated neurocognitive disorder (HAND) [10]. In chronic infection, the CNS can be a site of locally replicating, compartmentalized HIV-1 infection, with persistence of unique viral quasispecies and even independent viral evolution [11]. CNS compartmentalization is prevalent in individuals with HIV-associated dementia, the most severe form of HAND that is now rarely seen with the advent of ART [12,13]. This has led to the premise that either local HIV-1 replication itself or the attendant CNS local inflammation may contribute to HAND pathogenesis. However, discordant viral populations between blood and CSF have been detected in neurologically asymptomatic individuals and in those with mild forms of HAND [14].

It is unclear how early, and by what processes, compartmentalized HIV-1 replication begins in the CNS compartment. Compartmentalized HIV-1 variants have been detected within the first two years of life in children infected through vertical transmission, with some cases of compartmentalization appearing to arise from stochastic events of sequestration of a single variant of multiple transmitted/founder (T/F) viruses within the CNS compartment, while others develop more gradually with local evolution over time [15]. In a study of adults with early but not acute infection, subpopulations of HIV-1 in CSF were compartmentalized during the first year of infection, with CSF variants longitudinally observed to evolve independently from blood, beginning as early as four months after estimated HIV-1 infection [16]. Finally, sequencing of CSF and blood variants during suppressive ART suggests that treatment initiated in the first months after HIV-1 acquisition may not prevent the establishment of HIV-1 compartmentalization in the CNS [17].

Given that CNS-compartmentalized HIV-1 has been identified early in the course of HIV-1 infection and that rapid shifts of HIV-1 quasispecies during AHI in individuals with multiple T/F viruses have been observed [18], it is possible that unique founder HIV-1 quasispecies are introduced to the CNS during initial transmission to this compartment, leading to establishment of localized HIV-1 subpopulations. Alternately, identical viruses may disseminate to all compartments during acute infection, with local CNS replication of HIV-1 or multiple ongoing CNS transmission events leading to the eventual emergence of unique CNS HIV-1 variants in some individuals. To date, limited studies have examined the genetic characteristics of HIV-1 in any extra-immune tissue compartment during AHI to understand the patterns of the earliest stages of viral tissue dissemination occurring in humans [19]. Here, we report the examination of HIV-1 sequences from blood and CSF samples obtained from 18 ART-naïve Thai participants infected with single or multiple CRF01_AE and CRF01_AE/B recombinant T/F viruses, which were identified during Fiebig stages II–V of AHI. We assessed CNS compartmentalization through single-genome amplification (SGA), next-generation sequencing (NGS), and phylogenetic analyses of founder viruses in CSF and plasma.

## 2. Materials and Methods

### 2.1. Study Participants and Sample Collection

Study participants with AHI were selected from the RV254 cohort. RV254 (or SEARCH 010, clinicaltrials.gov NCT00796146) is a longitudinal observational study initiated in 2009 by the United States Military HIV Research Program, South-East Asian Research Collaboration (SEARCH), and the Thai Red Cross Anonymous Clinic in Bangkok, Thailand, on HIV to identify and characterize AHI with the intent of defining the earliest systemic virological and immunological aspects of HIV-1. Participants were offered ART at diagnosis of HIV infection. The study was approved by the Institutional Review Boards of the Walter Reed Army Institute of Research (Silver Spring, MD, USA), Chulalongkorn University (Bangkok, Thailand), and Yale University (New Haven, CO, USA). All participants underwent HIV testing by pooled nucleic acid methodology or sequential immunoassay. The duration of exposure was estimated from a detailed questionnaire on sexual history. Plasma and CSF samples were collected at baseline prior to ART initiation. All individuals were antiretroviral-naïve at the time of baseline study sampling. To facilitate viral amplification and sequencing, we selected pre-ART AHI samples in which both CSF and plasma HIV-1 RNA levels were greater than 10,000 copies/mL. Deidentification of participants’ study number and sequence name was performed for confidentiality.

The Sanger sequences obtained from this study are available in GenBank under accession numbers MK272339-MK272735, KT185688-KT185700, KT185711-KT185723, KT185754-KT185766, KT185780-KT185792, KT185803-KT185821, KT185832-KT185850, KT185861-KT185893, KT185904-KT185920, KT186015-KT186028, KT186039-KT186052, MG989497, MG989499, MG989502, MG989512, MG989519, MG989521, MG989527, MG989530, MG989537, MG989538, MG989543, MG989554, MG989566, MG989568, MG989569, MG989591, MG989593, MG989608, MG989640, MG989642, and MG989658. The next-generation sequencing data obtained from this study are available in the NCBI Sequence Read Archive (SRA) SRR8280871-SRR8280987, under BioProject PRJNA507242 and Biosample SAMN10485692-SAMN10485796.

### 2.2. Laboratory Methods

The study was performed in two steps to conserve the limited available volume of CSF. The first step utilized SGA sequence analysis of full genome or two half-genome equivalents from plasma and CSF of the first 11 participants. Additional envelope sequences were obtained for some of these participants. The SGA sequence data were used to determine subtype, calculate genetic diversity, and identify single or multiple T/F viruses. The second step deployed next-generation sequencing platforms with analysis pipelines to examine the proportion of virus variants in the two compartments. This step included sequences from 10 of the first 11 participants (one participant, 2545573, was excluded due to insufficient CSF). Initial analyses indicated evidence of compartmentalization in participants with multiple T/F viruses but not in those with single T/F viruses. We obtained specimens from an additional seven participants with preliminary indication of multiple T/F infection and similarly applied the two study steps: analysis of SGA genomes (full or two half-genome equivalents) from plasma only, followed by NGS of plasma and CSF, during which five of these seven participants were confirmed to have multiple T/F infections.

#### 2.2.1. Single-Genome Amplification and Sequencing of HIV-1 Viruses

HIV-1 RNA extracted from plasma and CSF using a QIAamp Viral RNA Mini kit (Qiagen, Germantown, MD, USA) was used as template to retrieve HIV-1 sequences either as a near full-length genome or as two half-genomes overlapping by approximately 1.5 kb. Briefly, viruses were pelleted by centrifugation and, when necessary, resuspended with phosphate-buffered saline without calcium and magnesium (Gibco, Grand Island, NE, USA) to obtain a final volume of 140 μL prior to extraction. cDNA was synthesized using SuperScript III First-Strand Synthesis System for RT-PCR (Invitrogen, Carlsbad, CA, USA) following the manufacturer’s protocol. A nested PCR with SGA strategy was employed to amplify cDNA at the dilution, yielding not more than 30% positive PCR products, as previously described [20]. A full genome or a half-genome nested PCR was performed using the Advantage GC Genomic LA Polymerase Mix kit (Clontech Laboratories, Inc., Mountain View, CA, USA) or the Expand High-Fidelity PCR System (Millipore Sigma, Burlington, MA, USA) following the manufacturer’s protocol. Primers used for cDNA synthesis and SGA amplification are shown in Appendix A. PCR products were purified and sequenced using an Applied Biosystems 3730 DNA Analyzer (Thermo Fisher Scientific, Foster City, CA, USA).

#### 2.2.2. Targeted Genome Region Next-Generation Sequencing

Subgenome regions in *pol* and *env* were selected for targeted deep sequencing (TDS). For both regions, cDNA was synthesized from plasma or CSF RNA and subjected to PCR amplification. A total of 2000 copies of cDNA were distributed into separate tubes for all except two CSF samples (1900 copies for participant 2536083 and 1400 copies for participant 2546340 due to low CSF specimen volume) at less than 200 copies to avoid PCR saturation. TDS of *pol* was performed using the Ion Torrent platform (Life Technologies, Carlsbad, CA, USA) with library preparation and sequencing methods as previously described [21]. TDS of *env* was performed using the PacBio platform (Pacific Biosciences, Menlo Park, CA, USA) with library preparation and sequencing chemistry 2.0 and 2.1, following the Procedure & Checklist - Preparing Amplicon Libraries using PacBio Barcoded Adapters for Multiplex SMRT Sequencing protocol for <1 kb fragment as provided by the manufacturer. The primers used for each genome region are described in Appendix A.

Most of the primers were designed based on consensus sequences of CRF01_AE and subtype B available in the Los Alamos National Laboratory (LANL) HIV database [22], and they were used as universal primers for amplification. However, specific primers were tailored for some participants based on SGA sequences from their plasma and CSF. A larger region of *pol* (HXB2 1817-3520) was amplified in the first round, and protease (PR, HXB2 2077-2601) and reverse transcriptase (RT, HXB2 2584-3322) were separately amplified in the second round. In addition, the V2 region of *env* (HXB2 6555-6978) was amplified. Hence, there were three HIV-1 genome regions examined in the study.

### 2.3. Data Analysis

#### 2.3.1. Single-Genome Amplification-Derived Sequence Analysis

A multiple alignment of reference strains and sequences of interest were generated using the MAFFT method in HIVAlign [22] and Gene Cutter [22] and were manually edited using Geneious Pro 5.5.5 (Biomatters, Auckland, New Zealand). Highlighter plots were created to visualize matches, mismatches, transitions/transversions, and silent/nonsilent mutations indicating genetic diversity of SGA-derived sequences [23]. An initial HIV-1 genotype was assigned for each sequence in this study using the online NCBI HIV-1 genotyping tool [24]. Molecular Evolutionary Genetics Analysis (MEGA) 5.0 [25] was used to perform phylogenetic analyses and generate maximum likelihood (ML) trees of the sequences of interest and reference HIV-1 subtype strains to designate subtype as well as determine sequence diversity and their relatedness within and between plasma and CSF compartments. Genetic distances were calculated using the Kimura 2-parameter model in MEGA.

#### 2.3.2. Next-Generation Sequencing Data Analysis

• Ion Torrent Platform

The sequence management and analyses to identify haplotypes were performed as previously described [21]. Briefly, fastq files were exported from the Personal Genome Machine (PGM) using Torrent Suite 4.4 software (LifeTechnologies, Carlsbad, CA, USA) and were subjected to quality control and alignment. For each alignment, the corresponding T/F sequence derived from Sanger sequencing was used as a reference. Quality control of Sequence Alignment/Map (SAM) alignments was performed using Samstat 1.08 [26]. Nautilus [27] was used to analyze SAM files in order to tally the frequency of each base at each sequence position and to determine the frequency of haplotypes. Alignments were also visualized using Tablet 1.16.09.06 [28]. The minimum coverage of 50,000 reads per base position, the comparable numbers of bidirectional sequencing, and the lower limit of quantification for single nucleotide substitutions at 0.5% were the criteria used in the analysis. The reproducibility of Ion Torrent sequencing was obtained by three independent replications of three samples, as seen in Appendix A.

• PacBio Platform

Raw data was analyzed using SMRTLink application versions 4.0 and 5.1 (Pacific Biosciences, Menlo Park, CA, USA). The subreads were demultiplexed, and circular consensus sequences (CCS) were generated for each barcode. Most of the default CCS parameters were used with some modifications on minimum number of passes = 5, minimum predicted accuracy = 0.995, and barcode score ≥65. The fastq file of CCS reads was aligned to the corresponding T/F sequence derived from Sanger sequencing as a reference using CoNvex Gap-cost alignMents for Long Reads (NGMLR) with default parameters [29]. SAM alignments were filtered for mapping quality (MAPQ) score ≥60 with soft-clip ≤75. A minimum coverage of 6000 reads was used to provide reliable detection of minor HIV-1 variant at 1%, as recommended by Pacific Biosciences [30]. The alignment was visually inspected for accuracy using Tablet 1.16.09.06 [28] and Sequencher 5.4.6 (Gene Codes Corporation, Ann Arbor, MI, USA) and subjected to haplotype analysis for either single T/F or multiple T/F infections.

• Haplotype Analysis for Single T/F Infection

For single T/F infection, a refined alignment was first performed. The mapped reads were used as input to Nautilus [27] to generate an alignment file, a count file of tabulated nucleotide frequency at each base position, and an insertion file containing parts of sequences to be refined. A Python script automating the MAFFT aligner was used to realign the nucleotides in the insertion file. The realigned nucleotides were inserted back into the main alignment file, from which the consensus sequence was generated. The refined alignment file was then reprocessed by Nautilus [27] to generate a new count file. The haplotype analysis was then performed using Nautilus version 2 beta (HJF, Bethesda, MD, USA), which extends functions to include analyses of indels and haplotype frequencies, and binning of haplotypes with corresponding SGA sequences. A position was defined as a haplotype position if the nucleotide variation was greater than 0.5% among the total reads and the difference between reverse and forward reads at that position was less than 5%. Any position with deletions or insertions associated with a homopolymer area was discarded. The consensus sequence of each haplotype (>0.5%) was generated and aligned with the corresponding SGA reference sequences by multiple sequence alignment. The final alignment was visualized using highlighter plot to identify presence of variations/mutations enriched in plasma or CSF.

• Haplotype Analysis for Multiple T/F Infection

For multiple T/F infections, the reads were mapped against the SGA reference sequences representing virus variants within infected individuals using Burrows–Wheeler Aligner (BWA) with parameter option for PacBio reads “bwa mem -x pacbio” [31]. These different primary alignments were combined as a SAM file, and all sequences were then converted to fasta format using Geneious Pro 5.5.5 (Biomatters, Auckland, New Zealand). The trimming script was used to trim primers or soft-clipping fragments upstream of the 5′- and downstream of the 3′-ends of the SGA reference sequences. Levenshtein distances of sequences within each primary alignment were calculated using the calculate-edit-distance script to obtain an edit distance (d) between 0 and 13 for each read and its reference. Sequences with an edit distance value of greater than 13 were classified as “others”. These criteria to identify the best cut-off point of edit distances were established from numerous experimental phylogenetic preanalyses and visual inspections of the subsampling sequences in each alignment. Briefly, the alignment file including subsampling sequences of “other” reads and all SGA references were used to generate the neighbor-joining tree using the Kimura 2-parameter model and complete deletion for gaps/missing data treatment [32]. These sequences were visually inspected using Geneious Pro 5.5.5 (Biomatters, Auckland, New Zealand) and Sequencher 5.4.6 (Gene Codes Corporation, Ann Arbor, MI, USA) to ensure correct classification.

• Statistical Analysis

Demographic and HIV disease-related characteristics are described as medians (minimum–maximum range). The time from the last self-reported HIV exposure was estimated for each subject. The median time was used for participants with multiple dates of possible HIV exposure. Mann–Whitney was used to compare continuous variables between groups. Analyses were performed with Prism 7.0a (Graphpad Software, San Diego, CA, USA).

## 3. Results

### 3.1. Study Participants

Eighteen participants (Fiebig stages II–V) were selected from RV254/SEARCH 010, an ongoing prospective study of acute HIV-1. The participants included 16 men and two women with a median age of 29 years (range 21–45) and a median CD4^+^ T cell count of 281 cells/mL (range 132–621). Median (range) of plasma and CSF HIV-1 RNA were 6.66 (5.07–7.75) and 5.13 (4.04–6.61) log10 copies/mL, respectively, and viral load in plasma was significantly higher than in CSF (*p* < 0.0001, paired *t*-test) (Table 1). Participants donated plasma samples at a median of 22 days post estimated HIV-1 exposure (range 14–32); CSF samples were collected within a median of one day (range 0–9) of plasma collection (Table 1).

### 3.2. Single-Genome Amplification and Sequencing Analysis

In total, 8–12 full genomes or two half-genome equivalent sequences were retrieved from plasma of all participants, and 10–11 two half-genome equivalent sequences were obtained from CSF of 10 out of 18 participants due to limited sample volumes. Additional envelope gene sequences (13–33 per individual) were generated from plasma and CSF of five participants to increase the power of the analysis. A total of 150 full genomes were retrieved from plasma, including 31 5′ half-genomes, 35 3′ half-genomes, and 76 envelope genes. A total of 101 5′ half-genomes, 101 3′ half-genomes, and 92 envelope genes were retrieved from CSF. The envelope sequences were mainly used in the analysis as it is well documented that *env* is the most phylogenetically informative region of the genome [33] (Table 1).

Phylogenetic analysis of plasma sequences using the NCBI HIV-1 genotyping tool revealed that most of the participants were infected with CRF01_AE, except one with a CRF01_AE and subtype B recombinant. ML trees of 5′ and 3′ half-genomes and reference subtype sequences were constructed (Appendix A). All of the CRF01_AE sequences clustered together with reference CRF01_AE sequences at significant bootstrap values of 99 and 100, further confirming the preliminary subtype designation by the HIV-1 genotyping tool. The genome structure of the sole recombinant strain harbored by participant 2544636 revealed that most of the envelope gene consisted of a CRF01_AE element (Appendix A). A phylogenetic tree of envelope gene sequences was constructed (Figure 1). Detailed analyses of available envelope sequences from plasma and CSF from each participant using highlighter plots and phylogenetic trees (Appendix A) revealed that 10 were infected with a single T/F virus and eight were infected with multiple T/F viruses (Table 1). Envelope sequences derived from plasma alone or plasma and CSF of participants with single T/F infection had high degrees of sequence similarity. In particular, participant 2544636 was infected with 100% homogenous quasispecies, and hence a phylogenetic tree and highlighter plot for this participant could not be generated.

The participants with multiple T/F infection had different patterns of diversity, indicating dissimilar proportions of virus variant distribution. Among the three multiple HIV-1 T/F-infected participants with SGA envelope sequences available from both CSF and blood, participant 2543625 only had the minor variant identified in plasma, while the major variants were found in both plasma and CSF. Participant 2546609 had two different variants as well as their recombinant forms identified. Major, minor, and recombinant variants were identified in plasma, while only the major and recombinants were found in CSF. The diversity among major variants was higher in CSF than in plasma. Participant 2546340 also harbored two different variants and their recombinant forms. However, the minor and recombinant viruses were only detected in plasma, while the major variant was found in both compartments. The results from highlighter plots and informative site analysis did not reveal any nucleotide signatures differentiating the sequences from the two compartments. For the other five participants with multiple T/F infections, SGA sequences were obtained only from plasma due to limited CSF volumes. Most of them (2547279, 2545383, 2549183, and 2547268) had minor variants between 10% and 20%. In participant 2548105, the proportion of the major variant was 50%, and the different minor variants were identified substantially.

Sequence diversity calculation was performed on 10 participants who had sequences from both plasma and CSF. The mean genetic pairwise distance of envelope sequences in the plasma compartment of the seven single T/F-infected participants ranged from 0.010% to 0.063%, whereas the mean genetic distances of the CSF compartment ranged from 0.023% to 0.102% (Table 1). The genetic diversity in these seven individuals, including the inter- and intracompartment sequences, ranged from 0.022% to 0.082%. The degree of sequence diversity within plasma and CSF from five of these single T/F-infected participants (2545573, 2546083, 2549583, 2543733, and 2549844) was significantly different: *p*-value < 0.001–0.026, with slightly higher diversity in CSF in most participants with the exception of 2549583. Sequences from participants 2543832 and 2545436 did not show significant differences between the two compartments. These data suggest that, in some participants, the genetic diversity in CSF was generally higher than in plasma. In three individuals infected with multiple T/F viruses (2543625, 2546609, and 2546340), the intraplasma compartment genetic distances ranged from 0.420% to 1.897%, and intra-CSF compartment ranged from 0.079% to 0.674%. The distances of sequences from plasma were significantly higher compared to those from CSF (each *p* ≤ 0.001). This may suggest some selective pressure that favors only virus variants in CNS. For eight individuals who did not have CSF envelope sequences, the plasma sequence diversity of three single infections were 0% to 0.054%, while the diversity of five multiply infected individuals ranged from 0.221% to 2.317%.

### 3.3. Next-Generation Sequencing

The results from SGA sequencing described above revealed a trend of higher genetic diversity in the CSF compartment versus blood for most participants infected with multiple T/F viruses. Hence, we employed next-generation sequencing platforms (Ion Torrent and PacBio) to assess the proportions of different variants across three HIV-1 genome regions in 17 out of 18 participants.

In the regions of the genome studied by Ion Torrent, a locus was determined within a window of 250 base pairs (bp) based on the reliable read length of sequences. Most of the regions sequenced using Ion Torrent had more than one locus identified, whereas only one locus was identified on PacBio-derived sequences with a read length between 350 and 650 bp. The haplotype position (nucleotide polymorphism position) included synonymous, nonsynonymous, and nucleotides not seen in SGA sequences. The variants in each locus were independent; i.e., variant 1 in locus 1 and in locus 2 were treated as distinct and might or might not have originated from the same virus quasispecies.

There was insufficient volume of CSF samples available to perform replicate assays; hence, we were unable to perform conventional statistical analysis to assess intercompartmental differences. Therefore, to demonstrate the reproducibility of the assay platforms, multiple T/F-infected participants were assayed in three independent replicates using Ion Torrent (Samples 1–3) and PacBio (Sample 4) platforms. The interassay coefficient of variation (CV) obtained from the two platforms (Appendix A) indicated that variants in excess of 3% for all three replicates had an assay CV of <15%. As there is no interassay gold standard for NGS variations reported in the literature, we focused on the variants present at greater than 3% in both compartments. By way of comparison, product literature for clinical HIV-1 tests based on qPCR lists the following acceptable precision ranges (log10 SD): 0.06–0.17 (Roche COBAS HIV-1 Test, version 2.0), 0.08–0.17 (Hologic Aptima HIV-1 Quant Dx Assay), and 0.09–0.30 (Abbott RealTime HIV-1).

The analysis using the criteria described revealed no differences between the two compartments in individuals with single T/F infections. However, for individuals with multiple T/F infections, two distinct patterns were observed. Two participants with multiple T/F infections (2547279, Figure 2, and 2546609, Appendix A) had similar proportions of HIV-1 major variants in both compartments, with relatively few minor variants identified. In contrast, the six other participants with multiple T/F infections (2543625, 2546340, 2545383, 2548105, 2549813, and 2547268) had comparatively more variants and compartmentalization between CSF and blood, as indicated by a greater than 1.5-fold difference of proportions of variants between compartments (Table 2). The analysis results of these multiply infected individuals were varied. Among specimens with the greatest sequencing coverage, results for three participants with differing degrees of compartmentalization (2547279, 2546340, and 2549813) are described below; results for the remainder of the participants can be found in Appendix A.

### 3.4. Participant 2547279

Plasma and matched CSF samples were collected at 28 days post estimated infection. Four genome loci (HXB2 position) were analyzed: POL PR I (2464-2486), POL RT I (2917-3011), POL RT II (3125-3271), and ENV V2 (6563-6959) (Figure 2). There were no differences in the proportions of major variants in plasma and CSF in any of the genome regions analyzed. One minor variant in the ENV V2 locus was present at 2.32-fold greater proportion in the CSF compared to plasma. However, proportions of minor variants were similar but slightly enriched in the CSF compartment among the other three loci, ranging from 1.04- to 1.47-fold difference.

### 3.5. Participant 2546340

Plasma and CSF were collected at 22 days and 23 days post estimated infection, respectively. Three loci were identified in POL PR (HXB2 position): POL PR I (2141-2248), POL PR II (2273-2390), and POL PR III (2417-2529). The variants and their proportions are shown in Figure 2. There were no differences in the proportion of major variants in the POL PR region, but minor variants (>3%) were found at a higher proportion in plasma versus CSF, with the fold differences ranging from 1.15 to 1.71. Three loci were identified in POL RT (HXB2 position): POL RT I (2656-2810), POL RT II (2843-3020), and POL RT III (3084-3248). The fold differences of major variants across all loci ranged from 1.19 to 1.22, and those of the minor variants ranged from 1.76 to 2.0. The proportions of minor variants were lower in CSF compared to plasma, similar to the results in the POL PR loci. There were no differences between the major and minor variants in the ENV V2 locus (HXB2 position 6468-6958).

### 3.6. Participant 2549813

Plasma and CSF were collected at 22 days and 23 days post estimated infection, respectively. There were three studied genome regions: POL PR, POL RT, and ENV V2. The variants and their proportions are shown in Figure 2. There were two loci identified in POL PR (HXB2 position): POL PR I (2124-2285), and POL PR II (2437-2444). The major variants in the first and second loci were present at 1.9- and 1.5-fold higher frequencies in plasma compared to CSF, respectively. In contrast, minor variants had higher proportions in CSF with fold differences ranging from 1.65 to 2.52. There were three loci identified in POL RT (HXB2 position): POL RT I (2612-2755), POL RT II (2858-2975), and POL RT III (3020-3197). The pattern of fold differences in POL PR was also observed in POL RT. All of the major variants in the three loci were higher in plasma, ranging from 1.5- to 1.68-fold higher, whereas the first minor variants identified over 20% were greater than twofold higher in CSF. Similar results were also seen in the ENV V2 locus (HXB2 position 6521-6959). The major variant was observed at a higher proportion in plasma with a fold difference of 1.69, which was in contrast to the higher proportion of the minor variant in CSF with a fold difference of 2.21.

In summary, the proportions of HIV-1 variants in plasma and CSF in participants with multiple T/F infections were highly variable between individuals. In two individuals (participants 2546609 and 2547279), differences in the proportions of variants between compartments were significant in a minority of sequenced regions and only among minor variants. In contrast, for participant 2549813, differences in variant proportions between compartments were significant in all sequenced regions and among major and minor variants. Several distinct patterns of distribution of major variants were observed, i.e., no difference in proportions between plasma and CSF for participants 2543625, 2546340, and 2545383; lower proportions in plasma for participants 2548105 and 2547268; or higher proportions in plasma for participant 2549813. Most minor variants had significantly different proportions in plasma and CSF. The proportions of minor variants were either lower in CSF, as seen in participants 2543625, 2546340, 2545383, 2547268, and 2548105, or higher, as seen in participant 2549813. These results indicate that certain variants are compartmentalized in AHI, differentially favoring either peripheral or CNS circulation.

## 4. Discussion

Our study provides the first data on the genetic relationships between plasma founder viruses and those disseminating to the CNS tissue compartment during the acute stage of HIV-1 infection. Prior reports have shown that HIV-1 can be detected in extra-immune tissue compartments, including CSF and brain, during the early stages of HIV-1 infection [34,35]. One study, also within the RV254/SEARCH 010 cohort, demonstrated extremely early detection of HIV-1 RNA in CSF at an estimated eight days post infection [2]. Compartmentalization of CSF-derived HIV-1 *env* has been detected as early as four months after infection in a separate study [16]. However, to examine whether the T/F viruses in the CNS are distinct from those in the blood during AHI, and to understand transmission dynamics between compartments, we undertook genetic examination of paired CNS and blood plasma samples during early AHI. Due to unique access to tissue samples from individuals during Fiebig Stage II–V of AHI, obtained at a median time of 22 days following estimated HIV-1 exposure, we were able to use SGA to demonstrate that viruses initially trafficking to the CNS compartment in humans are highly similar in sequence and genetic diversity to those contemporaneously found in plasma. This similarity between HIV-1 sequences in the CSF and the plasma might be expected because the systemic circulation is the likely source of virus. However, models for HIV-1 trafficking into tissues outside of the blood compartment often suggest mechanisms that may lead to an initial genetic bottleneck or selection of HIV-1 variants, evidence of which may be rarely observed via the standard SGA approach due to low sampling depth (approximately 30 viral genomes). Taking the further step of performing in-depth sequencing and analysis of the viruses using next-generation sequencing platforms (sampling 1400–2000 viral genomes), we did not see any differences between the CSF and blood compartments in individuals with single T/F infections. However, in all participants with multiple T/F infections, we found differences in the proportions of HIV-1 variants between the two compartments. The differences were highly variable between individuals. In some participants, the proportions of major HIV-1 variants were similar in both compartments, while they were lower or higher in plasma compared to CSF in other participants. Interestingly, the proportions of minor variants were different between compartments in most individuals with multiple T/F viruses. This implies that unknown mechanisms for HIV-1 trafficking into tissues outside of the blood compartment may lead to an initial genetic bottleneck or selection of HIV-1 variants at the very early stage of infection. Ultimately, this selection pressure may lead to the initial establishment of selected forms of HIV-1 variants in the CNS.

Data suggests that HIV-1 may be carried into the CNS through transmigration of infected immune cells across the blood–CNS barriers or meningeal lymphatic structures rather than through passage of free virus [36,37]. A study demonstrated that natalizumab, an anti-alpha-4 integrin antibody that prevents trafficking of leukocytes across the endothelial layers of the blood–CNS barriers, prevents integration of HIV-1 DNA in the brains of macaques, strongly supporting this concept [38]. Finding identical HIV-1 sequences in CSF and blood in single T/F-infected individuals with AHI supports either traffic of free virus or traffic of CD4^+^ T lymphocytes or monocytes with subsequent local CSF release of HIV-1 RNA. Many models have suggested that monocytes may play a key role in trafficking HIV-1 to the CNS, with subsequent establishment as perivascular macrophages acting as resident infected cells within the CNS [10]. The current study found enrichment of minor variants in CSF, possibly reflecting HIV-1 adaptation to preferentially enter and infect myeloid lineage cells [11,39]. Similarly, previous reports have shown characteristic genetic signatures associated with the CNS compartment in CSF- or brain-derived HIV-1 sequences [40] or compartmentalized unique CNS-derived sequences that may specifically associate with HAND [41]. One potential explanation for this is that viruses selectively gain access to the CNS compartment due to specific neurotropism or cell entry requirements.

Recently, our study on differential infection of cultured peripheral and CNS cells by distinct T/F HIV-1 infectious molecular clones (IMC) revealed that T/F variants in AHI show different efficiencies of replication in CNS cells (microglia and astrocytes) compared to peripheral cells (CD4^+^ cells and macrophages) in vitro and that p24 production by these variants is differentially influenced by TNFα [42]. Preferentially infected CNS-resident microglia, macrophages, and astrocytes may thus contribute to reservoir establishment of CNS-adapted HIV-1 variant(s) and subsequent compartmentalization in the brain. Additionally, based on our findings, wholly unique CSF sequences appear to develop later within the CNS. This would support evolution after initial viral CNS entry or later events associated with transport by trafficking cells.

The concept that viral compartmentalization occurs after initial distribution to tissues is congruent with existing knowledge of the factors influencing HIV-1 evolution. Immune responses to HIV-1 are a major determinant of diversity and envelope evolution [43]. The immune milieu in the CNS is characterized by a distinct collection of resident immune cells (e.g., microglia, tissue macrophages) and particular subsets of trafficking T cells and monocytes that possess an activated phenotype, facilitating their entry into this protected environment [44]. Furthermore, immunological studies of participants in the RV254/SEARCH 010 cohort have revealed a unique repertoire of HIV-specific T lymphocyte responses within the CSF compared to plasma during AHI, indicating distinct adaptive immune responses to HIV in the CNS starting very early during infection [45]. While HIV-1 that exhibits CNS compartmentalization in early infection appears to be adapted to replicate within T lymphocytes [16], a characteristic of late-stage or dementia-related compartmentalization is virus adaptation to replicate in cells with low plasma membrane CD4 density [11,39], suggesting macrophage tropism. Such forces (e.g., local immune effects, cellular targets for infection, and replication) would gradually select for virus with unique features adapted to this specific tissue compartment. Our study during AHI indicates that, following transmission of a single HIV-1 variant, HIV-1 detected in the CSF does not manifest any features, suggesting adaptation to the CNS, whereas the transmission of multiple HIV-1 variants results in compartmentalization of CNS variants. These findings are consistent with those in perinatally infected infants and children in Malawi, where enrichment or sequestration of variants in the CSF were observed in multiple T/F infections [15].

The sequences of HIV-1 variants found in plasma and in CSF are similar during AHI. With the standard SGA methods, there is no consistent evidence of compartmentalization except nonsignificantly higher diversity of CSF sequences because the capacity to detect the onset of CNS compartmentalization is limited by sampling depth to variants present at >10%. The inclusion of next-generation sequencing allowed us to address sampling depth [21] and revealed compartmentalization with respect to differences in proportions of HIV-1 variants between blood and CNS.

Although it is unknown how representative CSF samples are of the state within the brain parenchyma, CSF is in direct contact with the brain tissue at the level of the arachnoid and is the only CNS tissue readily available for analysis in living human study participants. Moreover, immune and viral characteristics of CSF have clearly revealed key aspects of HIV-1 neuropathogenesis, and compartmentalization of CSF-derived HIV-1 associates with encephalitis and neurological disease in HIV-1 [46]. We hypothesize that, early in the course of HIV-1 infection, the meninges and perhaps choroid plexus may be the primary site of HIV-1 replication within the CNS, whereas infection of cells within the brain parenchyma may typically occur at a later stage, facilitated by early persistent infection of the meninges and continued release of HIV-1 into the CSF.

This study of individuals identified within the first weeks following HIV-1 infection offers insights into the HIV-1 quasispecies initially entering the CNS. In the current study, individuals identified with AHI were offered immediate ART, opening the possibility that, although HIV-1 has entered multiple compartments at the time of treatment initiation, an absence of local compartmentalized infection may limit the persistence of HIV-1 in the CNS compartment over time in these early treated individuals.

## Figures and Tables

**Figure 1 cells-08-00902-f001:**
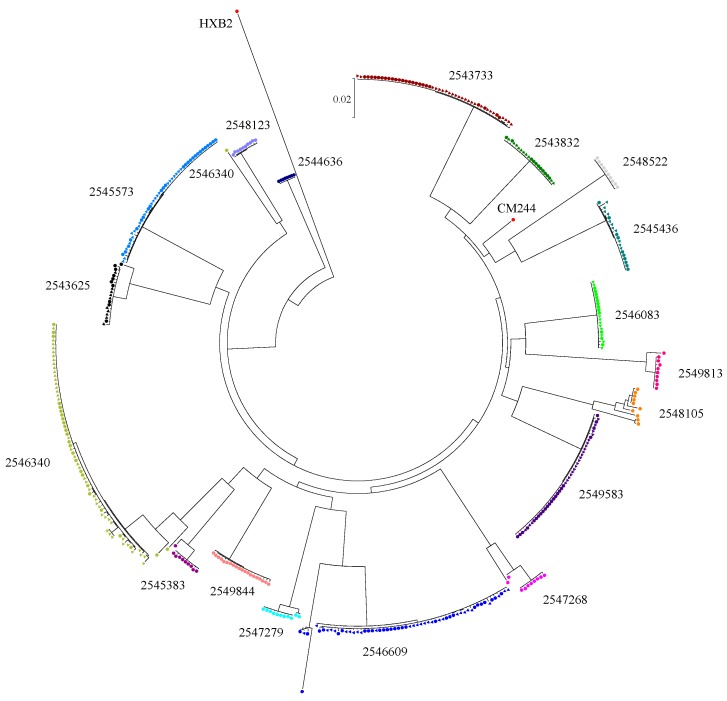
Maximum likelihood tree of envelope gp160 gene sequences. Sequences were retrieved from participant plasma (circle symbol) and cerebrospinal fluid (CSF) (triangle symbol). Reference sequences include HXB2 and CM244 (GenBank accession K03455 and AY713425, respectively).

**Figure 2 cells-08-00902-f002:**
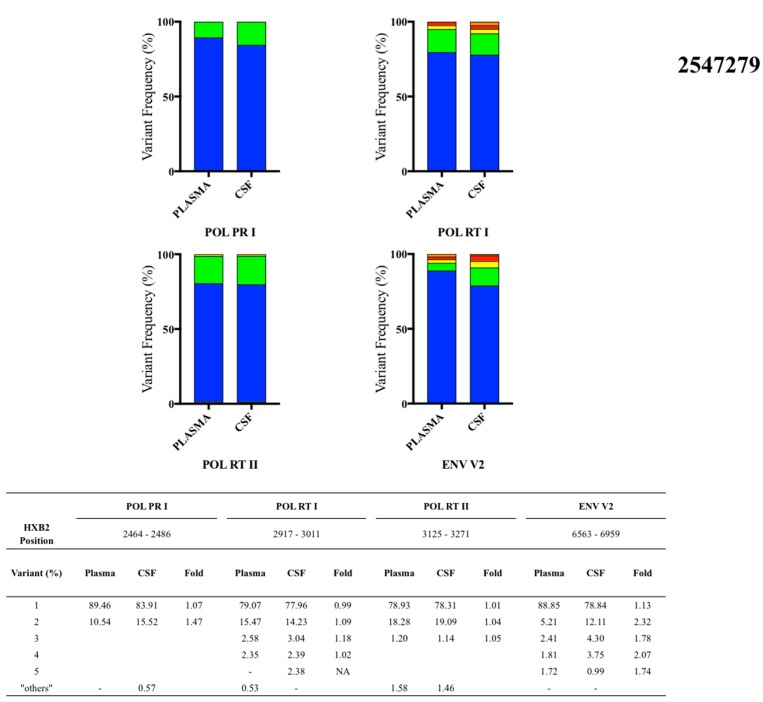
Proportions of plasma and CSF variants from three multiple transmitted/founder (T/F)-infected participants observed by next-generation sequencing platforms: participant 2547279, participant 2546340, and participant 2549813.

**Table 1 cells-08-00902-t001:** Demographic and clinical characteristics of participants, and genetic distances of envelope sequences from plasma and CSF.

Participant ID	Sex/Age (Years)	Transmitted/Founder (T/F) Infection	Subtype	Fiebig Stage ^1^	CD4 Cells/mL	Plasma HIV RNA log_10_ Copies/mL	CSF HIV RNA log_10_ Copies/mL	CSF WBC Cells/mL	Sample Time (Estimated Days Post Infection)		Number of Sequences (Plasma:CSF)	Env Genetic Diversity			
												Mean (SD)			P-Value ^2^
									Plasma	CSF		Intra Plasma	Intra CSF	Inter Plasma/CSF	Intra Plasma vs. Intra CSF
2545573	M/28	Single	CRF01_AE	II	426	6.50	4.13	10	14	15	26:23	0.062% (0.054%)	0.071% (0.053%)	0.066% (0.055%)	0.001
2546083	F/29	Single	CRF01_AE	IV	463	5.76	4.64	50	32	35	12:10	0.026% (0.033%)	0.047% (0.048%)	0.036% (0.043%)	0.026
2549583	M/30	Single	CRF01_AE	III	293	5.59	5.39	0	26	27	23:23	0.03% (0.03%)	0.023% (0.033%)	0.027% (0.032%)	0.002
2543832	M/36	Single	CRF01_AE	II	269	7.56	4.07	0	18	27	11:10	0.021% (0.032%)	0.023% (0.024%)	0.022% (0.03%)	0.25
2545436	M/32	Single	CRF01_AE	III	621	7.49	4.04	160	16	19	10:11	0.047% (0.035%)	0.062% (0.046%)	0.054% (0.042%)	0.22
2543625	M/21	Multiple	CRF01_AE	III	181	5.52	5.41	110	22	27	10:10	0.42% (0.615%)	0.079 % (0.079%)	0.252% (0.46%)	<0.0001
2543733	M/30	Single	CRF01_AE	II	569	5.07	4.64	4	15	16	24:24	0.01% (0.018%)	0.055% (0.045%)	0.033% (0.035%)	<0.0001
2549844	M/24	Single	CRF01_AE	V	231	6.12	5.20	7	16	19	10:10	0.063% (0.04%)	0.102% (0.067%)	0.082% (0.058%)	0.013
2546609	M/29	Multiple	CRF01_AE	III	206	6.61	4.41	3	28	29	29:29	1.897% (3.811%)	0.674% (2.131%)	1.286% (3.157%)	<0.0001
2546340	M/29	Multiple	CRF01_AE	IV	338	7.37	5.55	0	23	24	27:43	1.315% (3.175%)	0.191% (0.188%)	0.773% (2.366%)	0.0002
2548123	F/45	Single	CRF01_AE	III	132	7.41	5.3	0	22	23	10:NA	0.054% (0.033%)	NA	NA	NA
2547279	M/27	Multiple	CRF01_AE	IV	234	6.86	5.08	0	28	28	11:NA	0.221% (0.259%)	NA	NA	NA
2545383	M/37	Multiple	CRF01_AE	III	252	7.41	5.54	0	20	20	10:NA	0.301% (0.497%)	NA	NA	NA
2548105	M/40	Multiple	CRF01_AE	V	374	5.84	4.07	0	25	26	10:NA	2.317% (2.121%)	NA	NA	NA
2549813	M/22	Multiple	CRF01_AE	II	165	6.71	6.61	0	19	20	10:NA	0.374% (0.620%)	NA	NA	NA
2547268	M/23	Multiple	CRF01_AE	III	334	6.95	5.22	0	32	33	10:NA	1.263% (2.093%)	NA	NA	NA
2544636	M/29	Single	B/CRF01_AE	II	453	5.41	5.17	0	25	25	8:NA	0.000% (0.000%)	NA	NA	NA
2548522	M/29	Single	CRF01_AE	III	265	7.75	4.22	0	17	18	10:NA	0.024% (0.024%)	NA	NA	NA

^1^ Fiebig Stage: Fiebig II (17-23 days) RNA+/p24 Ag+/HIV IgM-; Fiebig III (20-27 days) RNA+/p24 Ag+/HIV IgM+/Western blot-; Fiebig IV (25-33 days) RNA+/p24 Ag+/HIV IgM+/Western blot indeterminate; Fiebig V (57-140 days) RNA+/p24 Ag+/HIV IgM+/Western blot+ but p31-; ^2^ Mann-Whitney Test; NA = Not available.

**Table 2 cells-08-00902-t002:** Compartmentalization of variant sequences in plasma and CSF.

	Compartmentalization of Variant Sequences (Ratio of Variants in Plasma:CSF > 1.5) *																			
	*pol protease*									*pol reverse transcriptase*									*env*	
	POL PR I			POL PR II			POL PR III			POL RT I			POL RT II			POL RT III			ENV V2	
Participant ID	Major	Minor		Major	Minor		Major	Minor		Major	Minor		Major	Minor		Major	Minor		Major	Minor
2546609	No	-		-	-		-	-		No	-		No	-		No	-		No	-
2547279†	No	No		-	-		-	-		No	No		No	No		-	-		No	Yes
2543625†	No	Yes		No	Yes		-	-		-	-		-	-		-	-		No	No
2545383†	-	-		-	-		-	-		No	Yes		No	Yes		-	-		No	Yes
2546340	No	No		No	Yes		No	Yes		No	Yes		No	Yes		No	Yes		No	No
2548105†	No	Yes		No	Yes		No	Yes		No	Yes		No	Yes		No	Yes		No	Yes
2547268†	No	Yes		No	Yes		-	-		No	Yes		No	Yes		-	-		No	Yes
2549813	Yes	Yes		Yes	Yes		-	-		Yes	Yes		Yes	Yes		Yes	Yes		Yes	Yes

* HXB2 coordinates of targeted deep sequencing regions available in figures for each participant; † Additional information available in Appendix A; - Insufficient data.

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
