# Peer review of "Deep Sequencing Reveals Central Nervous System Compartmentalization in Multiple Transmitted/Founder Virus Acute HIV-1 Infection"

_cells, 2019, doi:10.3390/cells8080902_

Round 1

Reviewer 1 Report

In Tovanabutra et al., is presented an in depth report of viral sequence diversity in paired plasma-CSF during acute HIV-1 infection with individuals who are being followed longitudinally. The authors have considered the role of selection.

There was no mention of a possible role for viral fitness and/or cooperativity in the context of multiple transmission, however this did not detract from the key message of the manuscript. 

Not sure that readers of Cell will be familiar with Fiebig Stage. 

Author Response

Dear Sir/Madam,

On behalf of the co-authors I am very grateful to you for reviewing and making very constructive comments that we have responded point-by-point as follows:

Reviewer comment: In Tovanabutra et al., is presented an in-depth report of viral sequence diversity in paired plasma CSF during acute HIV1 infection with individuals who are being followed longitudinally. The authors have considered the role of selection.

Response: The authors appreciate the reviewer’s comments and agree that this is a detailed study where we strive to consider factors including selection that may ultimately lead to CNS compartmentalization in multiple infected individuals.

Reviewer comment: There was no mention of a possible role for viral fitness and/or cooperativity in the context of multiple transmission, however this did not detract from the key message of the manuscript.

Response: The authors have further investigated viral fitness and infectivity of variants from plasma multiple transmitted founder viruses in multiple cell types (purified CD4+ T cells, cultured astrocytes and microglia, and peripheral macrophages). We found that infection with infectious molecular clones constructed from these variants resulted in sustained viral replication in all four cultured cell types. Preliminary results from this study were presented at the HIVR4P conference (Balingnang et al, 2019). We agree that further investigation of phenotypic and functional characteristics of these variants is important for further understanding of the pathogenesis and establishment of HIV in the CNS during acute HIV, but feel that this work is outside of the scope of the current manuscript.

Reviewer comment: Not sure that readers of Cell will be familiar with Fiebig Stage.

Response: We acknowledge that the readers of Cells may not be familiar with the definition of Fiebig stages. We have included the definition of Fiebig stages, as shown below in Table 1.

*Fiebig Stage:

Fiebig II (17-23 days) RNA+/p24 Ag+/HIV IgM-

Fiebig III (20-27 days) RNA+/p24 Ag+/HIV IgM+/Western blot-

Fiebig IV (25-33 days) RNA+/p24 Ag+/HIV IgM+/Western blot indeterminate

Fiebig V (57-140 days) RNA+/p24 Ag+/HIV IgM+/Western blot+ but p31-

The authors hope that our responses have been made clear to the reviewer.

Sincerely yours,

Sodsai Tovanabutra, PhD Chief, Viral Sequencing Core, HJF In support of the U.S. Military HIV Research Program (MHRP) Walter Reed Army Institute of Research (WRAIR) 503 Robert Grant Avenue, Room 2N25 Silver Spring, MD 20910 Phone: 301 319 9993 Cell: 301 339 3323 Email: [email protected]

Reviewer 2 Report

Deep Sequencing Reveals Central Nervous System Compartmentalization in Multiple Transmitted/Founder Virus Acute HIV-1 Infection

Authors Tovanabutra S et al. have examined the HIV-1 transmitted/founder (T/F) viruses from the central nervous system (CNS) and compared with the viruses from blood plasma collected at baseline prior to ART initiation. They used single genome sequencing of the envelope gene, and further examined sub-regions in pol and env using next generation sequencing (NGS) in paired plasma and CSF from 18 antiretroviral treatment (AR)-naïve Thai individuals infected with single or multiple CRF01_AE and CRF01_AE/B recombinant T/F viruses, identified during Fiebig stages II-V of AHI.

The results show the genetic relationships between plasma founder viruses

and those disseminating to the CNS tissue compartment during the acute stage of HIV-1 infection

The sequences of HIV-1 variants found in plasma and in CSF are similar during AHI. The authors demonstrate that using single genome amplification (SGA) that viruses initially reaching the CNS compartment in humans are highly similar in sequence and genetic diversity to those simultaneously found in plasma. Whereas the in-depth sequencing and NGS showed the differences.

The study is very interesting and useful for scientists working not only in the field of HIV but other RNA viruses too.

Line 114 – viruses, “who” should be changed to viruses “which”

Author Response

Dear Sir/Madam,

On behalf of the co-authors I am very grateful to you for reviewing and making very complementing comments that we have responded as follows:

Reviewer comment: The study is very interesting and useful for scientists working not only in the field of HIV but other RNA viruses too.

Response: Thank you very much for your time to review and make very encouraging comments on the scientific findings and the sequence methodology. The authors appreciate and agree with the view that the data and analysis presented in the manuscript will be useful for investigators in HIV as well as additional viruses.

Line 114 – viruses, “who” should be changed to viruses “which”

Response: We have corrected Line 114 as advised.

Sincerely yours

Sodsai Tovanabutra, PhD Chief, Viral Sequencing Core, HJF In support of the U.S. Military HIV Research Program (MHRP) Walter Reed Army Institute of Research (WRAIR) 503 Robert Grant Avenue, Room 2N25 Silver Spring, MD 20910 Phone: 301 319 9993 Cell: 301 339 3323 Email: [email protected]